# PIN2 Polarity Establishment in *Arabidopsis* in the Absence of an Intact Cytoskeleton

**DOI:** 10.3390/biom9060222

**Published:** 2019-06-07

**Authors:** Matouš Glanc, Matyáš Fendrych, Jiří Friml

**Affiliations:** 1Institute of Science and Technology Austria (IST Austria), 3400 Klosterneuburg, Austria; matous.glanc@psb.vib-ugent.be (M.G.); fendryc1@natur.cuni.cz (M.F.); 2Department of Experimental Plant Biology, Faculty of Science, Charles University, 12844 Prague, Czech Republic

**Keywords:** cell polarity, polarity establishment, PIN auxin efflux carriers, cytoskeleton, actin, microtubules, live-cell imaging

## Abstract

Cell polarity is crucial for the coordinated development of all multicellular organisms. In plants, this is exemplified by the PIN-FORMED (PIN) efflux carriers of the phytohormone auxin: The polar subcellular localization of the PINs is instructive to the directional intercellular auxin transport, and thus to a plethora of auxin-regulated growth and developmental processes. Despite its importance, the regulation of PIN polar subcellular localization remains poorly understood. Here, we have employed advanced live-cell imaging techniques to study the roles of microtubules and actin microfilaments in the establishment of apical polar localization of PIN2 in the epidermis of the *Arabidopsis* root meristem. We report that apical PIN2 polarity requires neither intact actin microfilaments nor microtubules, suggesting that the primary spatial cue for polar PIN distribution is likely independent of cytoskeleton-guided endomembrane trafficking.

## 1. Introduction

Auxin is arguably the single most important regulator of plant growth and development. Pivotal to its function are auxin distribution gradients, or local asymmetries in the levels of auxin signaling between cells. The auxin gradients, besides requiring local auxin biosynthesis, depend largely on polarized intercellular auxin transport, which is in turn mediated by multiple families of auxin transporters [1,2,3]. The directionality of auxin flow is determined by the polar subcellular localization of the PIN-FORMED (PIN) auxin efflux carriers [4]. The regulation of cell polarity in general, and PIN polarity in particular, is thus a crucial, yet still mostly unanswered, question in plant biology. PIN polar localization is thought to be regulated mainly at the level of PIN endomembrane trafficking [5]. The movement of endomembrane vesicles depends on the cytoskeleton, and accordingly, pharmacological interference with actin or microtubules (MTs) has been linked to defects in PIN subcellular dynamics [6,7]. However, opinions on the importance of the cytoskeleton for PIN polarity as such differ [7,8]. Here, we analyzed the effects of cytoskeleton-depolymerizing agents on the generation of apical PIN2 polarity in *Arabidopsis* roots using time-lapse imaging of advanced fluorescent reporters. We found, surprisingly, that the correctly polarized subcellular PIN distribution can be achieved in the absence of both intact actin and MTs. Our findings suggest that the primary PIN polarization mechanism is likely independent of cytoskeleton-mediated endomembrane trafficking.

## 2. Results

During cytokinesis of *Arabidopsis* root cells, PIN polarity is transiently lost and subsequently re-established [6,8,9,10]. We have recently shown that, in combination with the temporally-restricted expression of PIN2-GFP from the cytokinesis-specific *KNOLLE* promoter, the dividing cells can be used as a model to study PIN polarity establishment in real time [10]. Here, we employed this system together with the actin- and MT-depolymerizing drugs Latrunculin B (LatB) and Oryzalin (Ory), respectively, to test the importance of the cytoskeleton for the establishment of PIN polarity.

Application of either of the two drugs caused severe cytokinetic defects, ranging from division plane misorientation to complete cell plate (CP)-formation failure, which are consistent with both actin and MTs playing key roles in multiple steps of cell division [11]. This provided a good internal control which confirmed that the treatments indeed disrupted the respective cytoskeletal network (Figure 1a–c, Appendix A). In both cases, the KNOLLE::PIN2-GFP signal was partly mislocalized to abnormal endomembrane compartments, providing further evidence of the functionality of the drugs. In spite of the disturbed cytoskeleton, however, apical polarity of the plasma membrane (PM)-localized signal was in both cases eventually achieved, albeit with a considerable delay (Figure 1b,c,g, Appendix A). Notably, in case of the Ory treatment, the establishment of apical polarity was preceded by apolar distribution of the PM signal (Figure 1c, Appendix A).

We next applied both LatB and Ory at the same time. This co-treatment led to efficient depolymerization of both actin and MTs, as visualized by the respective markers Fimbrin-GFP and GFP-MAP4 (Figure 1e–f), and completely abolished the cells’ ability to divide (Figure 1d, Appendix A). Under these conditions, much of KNOLLE::PIN2-GFP localized to ectopic endomembrane aggregates that displayed only slow random movement and never reached the PM, indicating that PIN2-GFP trafficking was indeed severely disrupted (Figure 1d, Appendix A). Nevertheless, some of the signal did appear at the PM, presumably due to fusion events between the PM and vesicles that appeared in its proximity by Brownian motion. This PM signal was initially apolar, as in the case of the Ory treatment alone. Nonetheless, surprisingly, apical polarity of the PM-bound signal was eventually correctly established in most cases, albeit significantly later compared to the mock-treated control or either of the treatments alone (Figure 1d,g, Appendix A). This shows that the cellular mechanism that establishes apical PIN2 apical polarity is independent of the intact cytoskeletal network.

To validate our findings outside of the context of (unsuccessfully) dividing cells, we generated an *RPS5a::LOX:nls-mCherry:lox::PIN2-GFP* construct and introduced it into the *HS::CRE* background. In the resulting plants, a 1-hour heat shock led to the onset of Cre/Lox recombination-mediated expression of PIN2-GFP only in a few random and therefore often isolated cells, thus allowing us to observe the dynamic generation of PIN2 polarity in interphase cells as well (Figure 2a,b). Consistently with the previous results, the LatB + Ory cotreatment did not prevent apical PIN2-GFP polarity generation in this experiment either (Figure 2c).

## 3. Discussion

We have shown that that the application of cytoskeleton-interfering drugs Latrunculin B, Oryzalin, or both at the same time did not prevent the establishment of apical polarity of PIN2 in newly divided or interphase cells. The defects of the root growth rate, endomembrane trafficking, and cytokinesis caused by the drugs were consistent throughout the experiments, suggesting that the effects of the drugs were stable (Appendix A and data not shown). In all experiments presented, depolymerization of actin, MTs, or both at the same time led to partial PIN2-GFP mislocalization to abnormal endomembrane compartments. This again confirms that both cytoskeletal components are involved in PIN intracellular trafficking, as previously proposed [6,7,8]. On the other hand, the transient apolar PIN2-GFP localization preceded polarity establishment in LatB + Ory-treated cells only when expressed from the *KNOLLE* promoter, suggesting that it reflected the specific situation at the G2/M phase of the cell cycle and not the general behavior of PIN2. Moreover, the polar distribution of KNOLLE::PIN2-GFP appeared later under all treatments, demonstrating that cytoskeleton-guided trafficking contributes to the speed and/or efficiency of PIN polar targeting. Nevertheless, the spatial cue for PIN2-GFP polarity establishment was cytoskeleton-independent in both mitotic and interphase cells, supporting similar conclusions drawn in the past for PIN1 [8]. This suggests that actin- and MT-guided endomembrane trafficking stabilizes cell polarity by reinforcing asymmetrical distribution of polar cargoes, including the PINs. However, both actin and MTs act downstream of an unknown, cytoskeleton-independent polarity establishment mechanism.

Therefore, cytoskeleton-dependent polarized exocytosis, either secretory or recycling, cannot be the main mechanism of PIN polarity generation as hypothesized previously [12]. Both LatB and Ory affected the lateral diffusion of PIN1-GFP in BY-2 cells [13], but since we have previously shown that lateral diffusion does not play a major role in the re-establishment of PIN2 polarity under normal conditions [10], we assume that it is not a major PIN polarizing factor upon LatB and/or Ory treatment, either. It has been theoretically predicted and experimentally verified that clathrin-mediated endocytosis plays a key role in PIN polarity regulation [9,10,14,15,16,17,18]. An endocytosis-dependent, cytoskeleton-independent system based on differential rates of PIN endocytosis at different PM domains is therefore a prime candidate for the main mechanism of PIN polarity establishment. In the case of apically localized PIN2, such a mechanism might, in principle, work in two different ways, or their combination: A) Endocytosis machinery might be generally less active at the apical PM compared to the rest of the cell, or B) PIN2 molecules in the apical domain could be specifically “protected” from endocytosis by an unknown factor. The observation that the CLC-GFP signal is enhanced at the lateral PM domains has already indirectly hinted at mechanism A) [15], while the existence or nature of any factor specifically modulating PIN2 endocytosis in a domain-specific manner, as predicted in B), is so far elusive. It was proposed that NPH3-like proteins encoded by the *MAB4/MEL* gene family might possess this function, but the evidence presented to support this claim is far from conclusive [19].

## 4. Materials and Methods

Seeds were grown on ½ Murashige–Skoog medium with 1% sucrose and 1% agar, and grown in vitro for 4 days under long-day conditions. The transgenic lines *KNOLLE::PIN2-GFP* [10], *35S::AtFIM1-GFP* [20], and *35S:: GFP-MAP4* [21] were described previously. The *RPS5a::LOX:nls-mCherry:lox::PIN2-GFP* construct was generated using the Gateway cloning technology (Invitrogen) as follows: The *RPS5a* promoter fragment was cloned into pDonrP4-P1r; a fragment comprising a *LOX* site, *mCherry* with an N-terminal nuclear localization signal, a *NOS* terminator, and another *LOX* site was commercially synthesized (Eurofins Genomics) and cloned into pDonr221; the *PIN2-GFP* fragment was cloned into pDonr P2r-P3. All three entry clones were recombined into the pB7m34GW,0 destination vector, and the resulting expression clone was transformed into the *HS::CRE* [22] background using the floral dip method [23]. 20 µM LatrunculinB and 40 µM Oryzalin (both Sigma) were applied from 20 mM or 40 mM DMSO stocks, respectively, in small slices of agar medium, as described previously [10]. Heat-shock induction of CRE recombinase expression was performed by placing the seedlings, mounted in chambered coverslips (LabTek), into a 37 °C incubator for 1 h. *KNOLLE::PIN2-GFP* and *RPS5a::LOX:nls-mCherry:lox::PIN2-GFP* time-lapse imaging was performed on a vertically oriented Zeiss LSM700 confocal microscope, as described previously [10,24]. Imaging of *35S::AtFIM1-GFP* and *35S:: GFP-MAP4* controls was performed on inverted Zeiss LSM700 or LSM880 microscopes.

## 5. Conclusions

In conclusion, we have demonstrated that apical PIN2 polarity can be generated and maintained in the absence of intact actin microfilaments and microtubules in both mitotic and interphase cells, as was previously also shown in the case of the basal polarity of PIN1 [8]. These observations exclude any cytoskeleton-guided vesicular trafficking-based mechanism as the main driving force of PIN polarity establishment. On the other hand, the substantial evidence that endocytosis plays a central role in PIN polarity [9,10,14,15,16,17,18] necessitates that regulation of PIN endocytosis, and specifically its rates at different PM domains, be subjected to detailed analyses, as these will likely shed light on the still enigmatic mechanisms of cell polarity establishment in plants.

## Figures and Tables

**Figure 1 biomolecules-09-00222-f001:**
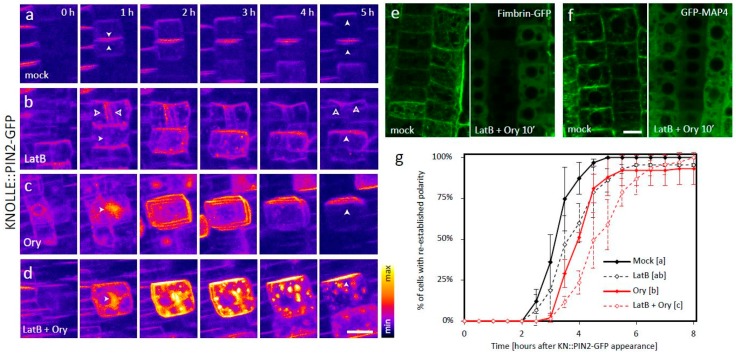
(**a**–**d**) Establishment of PIN2 polarity in root epidermal cells, visualized by dynamic localization of PIN2-GFP expressed from the mitosis-specific *KNOLLE* promoter. (**a**) In a control situation, the signal localizes initially to the CP-derived new PM pair, and the apical polar distribution typical for PIN2 is reestablished 2–5 h afterwards [10]. *n* = 51 cells from six roots in three independent experiments. (**b**) In LatB-treated cells, part of the signal localizes to ectopic endomembrane compartments, but the distribution dynamics of the PM signal are similar to the control if the CP is formed (upper cell; empty arrows). Apical polar distribution also develops in cells where the treatment leads to cytokinetic failure and the signal is initially found in ectopic endomembrane aggregates instead of the CP (lower cell; full arrows). *n* = 87 cells from 11 roots in three independent experiments. (**c**) In Ory-treated cells, the signal is initially localized to a ball-shaped endomembrane aggregate that forms instead of the cell plate. Most of the signal is subsequently relocalized to the PM initially in an apolar manner, followed by development of apical polarity. *n* = 90 cells from 11 roots in three independent experiments. (**d**) Cells treated with both inhibitors at the same time display strict cytokinetic failure and signal localization to large, mostly immobile endomembrane aggregates, in addition to the PM. Nevertheless, apical polarity of the PM-bound signal is eventually also achieved in this case. *n* = 83 cells from nine roots in three independent experiments. (**e**) and (**f**) The markers Fimbrin-GFP (**e**) and GFP-MAP4 (**f**) confirm that the LatB + Ory cotreatment leads to fast and efficient depolymerization of both actin and MTs, respectively. *n* = 19–20 roots from three independent experiments per line and treatment. (**g**) Quantification of the data presented in (**a**)–(**d**). The timepoint of KNOLLE::PIN2-GFP apical signal polarity establishment was marked for each cell, and the percentage of cells with established polarity in each condition was plotted against time. The mean +/− SD of three independent experiments, each consisting of >20 cells from 2–4 different roots (see *n* values in (**a**)–(**d**)), is shown. The letters [a–c] indicate statistical significance based on the Kruskal–Wallis Rank Sum Test (post hoc analysis: Alpha: 0.01, *p*-value adjustment: Bonferroni), and the data points from all three experimental replicates were pooled for the purpose of statistical analysis. A typical LatB + Ory treated cell, as well as a rare one where polarity re-establishment failed, are shown in Appendix A. Scale bars = 10 μm.

**Figure 2 biomolecules-09-00222-f002:**
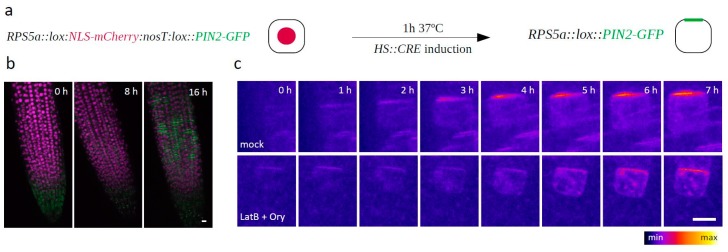
(**a**) Schematic depiction of the *RPS5a::lox:NLS-mCherry:nosT:lox::PIN2-GFP* construct generated to monitor PIN2-GFP polarity establishment in interphase cells. (**b**) Merged red and green channels of an *RPS5a::lox:NLS-mCherry:nosT:lox::PIN2-GFP*-expressing root, 0, 8, and 16 h after the heatshock induction of CRE recombinase expression. (**c**) Establishment of PIN2 polarity in Mock- and LatB + Ory-treated *RPS5a::lox:NLS-mCherry:nosT:lox::PIN2-GFP* cells. The green channel is shown; t0 = the last timepoint before the PIN2-GFP signal appeared. *n* > 30 cells from six different roots in two independent experiments. Scale bars = 10 μm.

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
