# Peer review of "PIN2 Polarity Establishment in Arabidopsis in the Absence of an Intact Cytoskeleton"

_biomolecules, 2019, doi:10.3390/biom9060222_

Round 1

Reviewer 1 Report

The manuscript entitled "PIN2 polarity establishment in Arabidopsis in the absence of intact cytoskeleton” presents the new data important for understanding one of the most important processes  related to the regulation of plant development, which is the subcellular polarization of PIN proteins, responsible for the directional transport of auxin. The authors show that the presence of a intact cytoskeleton is not essential for proper polarity of PIN2 proteins in the root epidermis and subsequently conclude that the  mechanism of  PIN  polarity establishment is probably based on differential rate of cytoskeleton-independent endocytosis.  The presented results are very interesting and important for the entire scientific community, and I have only few comments to the manuscript.

Lines 22-23. The statement "the primary spatial cue for polar PIN distribution is independent of cytoskeleton-guided endomembrane trafficking" is in my opinion too categorical for the current state of knowledge and available research methods. For example, visualization of the cytoskeleton using the indirect method of localization of binding proteins, which may or may not be connected to them cannot fully reflected the cytoskeleton state in terms of its functionality. Therefore this conclusion should be softened.

Line 20. „in the epidermis of the Arabidopsis root meristem”  - The cells in meristems are undifferentiated, so I have doubts whether the surface layer of the root meristem should be called the epidermis. For the root apex, the term "protoderm" would be more appropriate.

Line 21. "actin" should be replaced by "actin microfilaments"

Line 43. The word "that" has been repeated twice

Fig. 1. In the  panel "b" images are difficult to understand and require at least a more detailed description. They would be more clear if the top and bottom cells were marked, and arrows pointing to important details were present in all the images; the location of the arrows in the last image is misleading, it is not clear which regions are indicating.

Discussion. Discussion. The damage to the cytoskeleton could affect the mobility of proteins in the cell membrane. Thus, it is possible that the proper polar localization of the PIN2 proteins, re-establish after treatment with the actin- or/and MT-depolymerizing drugs, could be arisen as a result of lateral diffusion of PIN2 proteins in the cell membrane. Discussing this possibility in the manuscript would be interesting.

Author Response

The manuscript entitled "PIN2 polarity establishment in Arabidopsis in the absence of intact cytoskeleton” presents the new data important for understanding one of the most important processes  related to the regulation of plant development, which is the subcellular polarization of PIN proteins, responsible for the directional transport of auxin. The authors show that the presence of a intact cytoskeleton is not essential for proper polarity of PIN2 proteins in the root epidermis and subsequently conclude that the  mechanism of  PIN  polarity establishment is probably based on differential rate of cytoskeleton-independent endocytosis.  The presented results are very interesting and important for the entire scientific community, and I have only few comments to the manuscript.

We thank the reviewer for these comments.

Lines 22-23. The statement "the primary spatial cue for polar PIN distribution is independent of cytoskeleton-guided endomembrane trafficking" is in my opinion too categorical for the current state of knowledge and available research methods. For example, visualization of the cytoskeleton using the indirect method of localization of binding proteins, which may or may not be connected to them cannot fully reflected the cytoskeleton state in terms of its functionality. Therefore this conclusion should be softened.

While we believe the drugs and cytoskeleton binding protein-based reporters used are reliable (in case of microtubules, the experiment presented in Fig.1F was repeated also with the UBQ::TUA6-Venus reporter, in which the fluorescent tag is translationally fused directly to the tubulin monomer, with identical result), we agree that our conclusions are only as good as these tools. We have modified the last sentence of the abstract to soften the conclusion, so that it now reads:

„suggestingthat the primary spatial cue for polar PIN distribution is likely independent of cytoskeleton-guided endomembrane trafficking.“

(instead ofimplying that the primary spatial cue for polar PIN distribution is independent of cytoskeleton-guided endomembrane trafficking.“).

Accordingly, we have also replaced „imply“ by „suggest … likely“ in the last sentence of Introduction (from line 43 onwards)

Line 20. „in the epidermis of the Arabidopsis root meristem”  - The cells in meristems are undifferentiated, so I have doubts whether the surface layer of the root meristem should be called the epidermis. For the root apex, the term "protoderm" would be more appropriate.

The cells in the meristem are indeed not fully differentiated, however, the radial organization of the root is set very early on in the meristem, and cells of (to be) epidermis, cortex, endodermis express the differentiation markers already early on in the meristem. Importantly, in the seminal work of Dolan et al (Development, 1993) that sets the nomenclature for the organization of the Arabidopsis root, the ‘epidermis’ hidden still by the lateral root cap cells is called epidermis (see fig.1), and so we would like to keep calling these cells epidermis.

Line 21. "actin" should be replaced by "actin microfilaments"

We have modified the text accordingly.

Line 43. The word "that" has been repeated twice

We have have corrected the typo.

Fig. 1. In the  panel "b" images are difficult to understand and require at least a more detailed description. They would be more clear if the top and bottom cells were marked, and arrows pointing to important details were present in all the images; the location of the arrows in the last image is misleading, it is not clear which regions are indicating.

Thank you for pointing out the suboptimal clarity of this panel. In order to accommodate this request, and at the same time to maintain the unity in style between panels 1a-d and avoid having multiple arrows in all 24 frames shown, we have 1) added arrows indicating predominant signal localisation at t=1h in addition to t=5h in all panels; 2) distinguished between the two cells presented in panel 1b by using empty arrows for the upper cell and full arrows for the lower cell; and 3) improved the explanation of the observations in the figure legend.

Discussion. The damage to the cytoskeleton could affect the mobility of proteins in the cell membrane. Thus, it is possible that the proper polar localization of the PIN2 proteins, re-establish after treatment with the actin- or/and MT-depolymerizing drugs, could be arisen as a result of lateral diffusion of PIN2 proteins in the cell membrane. Discussing this possibility in the manuscript would be interesting.

Thank you for this comment. We added a sentence discussing this on lines 144-148 of the revised version of the manuscript. We have also included an extra reference to Laňková et al., 2016, where the effect of cytoskeleton-depolymerizing drugs on lateral diffusion of membrane proteins including PIN1 was studied in BY-2 cells, and re-numbered the subsequent references accordingly.

Reviewer 2 Report

The manuscript presented by Glanc et al. describes the novel finding regarding the regulation of apical polar localization of PIN2 in the epidermis of the Arabidopsis root meristem. Using advanced live-cell imaging techniques, the authors go on to suggest that apical PIN2 polarity requires neither intact actin nor microtubules, implying that the primary spatial cue for polar PIN distribution is independent of cytoskeleton-guided endomembrane trafficking.

The authors performed an elegant system where the dividing cells were used as a model to study PIN polarity establishment in real time together with pharmacological interference of actin or/and microtubules using the depolymerizing drugs, Latrunculin B and Oryzalin.

The authors demonstrated that apical PIN2 polarity could be generated and maintained in the absence of intact actin and MTs in both mitotic and interphase cells.

The work represents an important area of research, namely the influence of the cytoskeleton-guided vesicular trafficking as a mechanism for the establishment of PIN polarity. The findings are novel and will be of significance to researchers in the field.

Minor issues:

120 Replace othe by other.

Author Response

The manuscript presented by Glanc et al. describes the novel finding regarding the regulation of apical polar localization of PIN2 in the epidermis of the Arabidopsis root meristem. Using advanced live-cell imaging techniques, the authors go on to suggest that apical PIN2 polarity requires neither intact actin nor microtubules, implying that the primary spatial cue for polar PIN distribution is independent of cytoskeleton-guided endomembrane trafficking.

The authors performed an elegant system where the dividing cells were used as a model to study PIN polarity establishment in real time together with pharmacological interference of actin or/and microtubules using the depolymerizing drugs, Latrunculin B and Oryzalin.

The authors demonstrated that apical PIN2 polarity could be generated and maintained in the absence of intact actin and MTs in both mitotic and interphase cells. 

The work represents an important area of research, namely the influence of the cytoskeleton-guided vesicular trafficking as a mechanism for the establishment of PIN polarity. The findings are novel and will be of significance to researchers in the field. 

We thank the reviewer for these comments.

Minor issues:

120 Replace othe by other.

We have corrected the typo.